# FAK downregulation suppresses stem-like properties and migration of human colorectal cancer cells

**Chunyan Xu**[1,2], **Wenlu Zhang**[3☯]*, **Chengxia Liu**[1,2☯]*

**1** Department of Gastroenterology, Binzhou Medical University Hospital, Binzhou, Shandong, China,
**2** Institute of Digestive Diseases, Binzhou Medical University Hospital, Binzhou, Shandong, China,
**3** Department of Respiratory, Binzhou Medical University Hospital, Binzhou, Shandong, China

☯ These authors contributed equally to this work.
* zhangwenlu2022@163.com (WZ); phdlcx@163.com (CL)

**Data Availability Statement:** All relevant data are within the paper.

**Funding:** The Medical and Health Science and Technology Development Program of Shandong Province (No. 2019WS331). The funders and

## Abstract

Focal adhesion kinase (FAK) is a cytoplasmic protein tyrosine kinase, which is overexpressed in colorectal cancer cells. FAK could be activated by phosphorylation to participate in the transduction of multiple signaling pathways and self-renewal of cancer stem cells. Whether the downregulation of FAK inhibits the metastasis in colorectal cancer through the weakening of stem cell-like properties and its mechanisms has yet to be established. CD44, CD133, c-Myc, Nanog, and OCT4 were known to mark colorectal cancer stem cell properties. In this study, AKT inhibitor (MK-2206 2HCl) or FAK inhibitor (PF-562271) decreased the expression of stem cell markers (Nanog, OCT4, CD133, CD44, c-Myc) and spheroid formation in colorectal cancer. Moreover, FAK and AKT protein was shown to interact verified by co-immunoprecipitation. Furthermore, downregulation of FAK, transfected Lenti-FAK-EGFP-miR to colorectal cancer cells, reduced p-AKT but not AKT and decreased the expression of stem cell markers and spheroid formation in colorectal cancer. In conclusion, we demonstrated that downregulation of FAK inhibited stem cell-like properties and migration of colorectal cancer cells partly due to altered modulation of AKT phosphorylation by FAK.

## Introduction

During the initiation of metastasis, epithelial cancer cells undergo an epithelial-mesenchymal transition (EMT), and as a result become migratory and invasive mesenchymal-like cells while acquiring stem cell-like properties and therapy resistance [1,2]. Cancer stem cells are equipped with highly efficient DNA repair and effective redox tolerance systems in order to promote therapy resistance [3], and their number is increased outstandingly in cancers with cold immunity [4]. The prime time for medical intervention is the point when cancer stem cell-like cells start appearing. Early diagnosis of cancer-related symptoms is encouraged to understand and develop strategies for the treatment of colon cancer. It was known EMT is a feature of stem cell-like properties in tumor [2]. CD44, CD133, c-Myc, Nanog, OCT4 and SOX2 were

sponsors did not play a role in the study design, data collection and analysis, decision to publish, or preparation of the manuscript.

**Competing interests:** The authors have declared that no competing interests exist.

known to mark colorectal cancer stem cell properties [5–7]. The transcription factor, Twist1, is a marker of the EMT which regulates matrix metalloproteinase (MMP) expression and has been identified as the downstream target of CD44 [8]. CD44 and CD133 were increased during the EMT of colorectal cancer cells [9]. C-Myc is the gene most strongly associated with cancer and positively regulates the expression of SnaiL, a master regulator of the EMT [10,11]. Furthermore, EMT facilitates the generation of cancer stem cell-like properties for metastasis, but also for self-renewal properties required for initiating secondary tumors attributed to Nanog, OCT4 and SOX [12].

Our previous study showed that Focal adhesion kinase (FAK) promotes the metastasis of colorectal cancer by inducing the EMT mediated by the AKT phosphorylation [13]. FAK is a cytoplasmic tyrosine kinase, which is overexpressed in cancer cells. FAK could be activated by phosphorylation to participate in the transduction of multiple signaling pathways and self-renewal of cancer stem cells [14,15]. AKT was frequently dysregulated in tumors and played a pivotal role in tumor metastasis. Prevention of AKT phosphorylation is key to targeting cancer stem-like cells [16]. Recent studies have shown that the FAK/AKT pathway was activated in the process of obtaining stem cell-like properties of cancer cells [17,18]. Thus, we speculated that FAK may regulate AKT phosphorylation to promote the acquisition of stem cell-like properties and the migration of colorectal cancer.

The current study attempted to investigate that downregulation of FAK inhibits metastasis in colorectal cancer through the weakening of stem cell-like properties and its mechanisms. Expression of FAK or AKT was manipulated by transfection experiments and the use of specific inhibitors, and then the stem cell-like properties and migration of colorectal cancer cells were assessed.

## Materials and methods

### Ethics statement

The research Ethics Committee of the Binzhou Medical University Hospital reviewed that the research content, scope and methods involved in this project meet the relevant requirements of medical ethics.

### Cell culture and transfection

Human adenocarcinoma cell lines (American type culture collection, USA) were cultured in DMEM supplemented with 10% FBS and 100 U/ml penicillin/ streptomycin at 37˚C with 5% $CO_2$. As the previous study, the Lenti-FAK-EGFP-miR ($5x10^7$ TU/ml) (Shanghai R&S Bio-technology Co., Ltd., Shanghai, China) was introduced into RKO cells via lentiviral transfection [multiplicity of infection (MOI) = 30]. RKO cells were additionally transfected with the blank vector (Lenti-EGFP-miR; $2x10^8$ TU/ml) (negative control; NC) (Shanghai R&S Biotechnology Co., Ltd., Shanghai, China) as control (MOI = 30), and stably transfected cells were selected and used for subsequent experiments [13].

### Spheroid formation assay

For spheroid formation assay, $1×10^3$ cells were seeded onto non-adherent 6-well culture plates (Costar, Catalog no.: 3471; Corning Inc., NY) in stem cell medium composed of serum free DMEM/F 12 medium (Gibco, USA) containing 10 mg/l epidermal growth factor (EGF, AF 10015, PeproTech Inc., NJ), 10 mg/l basic fibroblast growth factor (basic-FGF, 10018 B, Pepro-Tech Inc., NJ) and 2% B27 supplement (50X, 17504044, Gibco, USA). Cells were incubated for 7 days, and spheroids with a diameter of >75 μm were counted under a light microscope.

## Western blotting analysis

Cells were collected by scraping and lysed with RIPA lysis buffer supplemented with 1% protease and 1% phosphatase inhibitor. Equal amounts of protein were separated by SDS–PAGE and blotted onto PVDF membranes. Membranes were scanned using an odyssey infrared imaging system (LI-COR, USA) and strips were analyzed with quantity one. Polyclonal antibodies against the following proteins were used: GAPDH (60004-1-1g,1:2000), FAK (12636-1-AP,1:2000), CD133 (18470-1-AP,1:1000), CD44(15675-1-AP, 1:2000), Nanog (14295-1-AP, 1:2000), Sox2(66411-1-1g, 1:2000), c-Myc(10828-1-AP, 1:1000) were from proteintech group, USA. Phospho-AKT (Ser473) (4060, 1:1000) and AKT (4691,1:1000) were purchased from cell signaling technology, USA. FAK (phosphor-Tyr397) (11215, 1:1000) was purchased from signalway antibody, USA.OCT4 (WL02020, 1:1000) was purchased from wanlcibio, China. Secondary IRDy® 680RD goat anti-rabbit (926–68071; 1:10000) and IRDy® 680RD goat anti-mouse antibodies (925–68070; 1:10000) were from LI-COR, USA.

## Wound healing assay

To assess wound healing, cells were seeded into a 6-well cell culture plate and when 80%-90% confluency had been achieved, the cell populations were scratched by a 1 mm spearhead. Images were acquired after 0 h and 24 h of incubation and migration distance was measured by microscope software (Nikon, Japan).

## Cell viability

Cell viability was estimated by CCK-8 assay kit (5007, Japan) according to the manufacturer's instructions. Absorbances were read by microplate reader (ELX800, USA) at 490 nm. FAK inhibitor (PF-562271) and AKT inhibitor (MK-2206 2HCl) were purchased from Selleck, China. Cell viabilities were expressed as percentages according to the following formula: (%) = A490 (sample)/A490 (control) × 100.

## Cell immunocytochemistry

Cells, at a density of $2x10^4$ cells/ml, were plated onto slides fixed in a culture dish, followed by treatment with MK-2206 2HCl or PF-562271 or 0.9% NaCl for 24 h. Slides were fixed with ice-cold 100% methanol, quenched with 0.3% $H_2O_2$ and blocked with normal goat serum. After incubation for 30 min with primary antibodies (as western blotting analysis) and washing, biotinylated secondary antibodies were added with incubation for 30 min, followed by washing and addition of preformed avidin/DH-biotinylated horseradish peroxidase H complex for 30 min. Slides were then overlain with DAB, rinsed, dried, mounted and a coverslip placed over them. Image pro plus analysis system was used to analyze protein expression. Three visual fields were randomly selected for each slide, and the total area and optical density were measured. Expression levels are expressed as average optical density (AOD): AOD = accumulated optical density/total area.

## Co-Immunoprecipitation

Protein A/G agarose beads (78609, Pierce, USA) were washed with PBS, and diluted to 50% with PBS. A 100 μL aliquot of Protein G Sepharose (50%) and 1 μL non-immune serum were added to 1 mL cell lysate and the supernatant was collected after centrifugation. 10 μg rabbit anti-FAK antibody was added to the experimental groups and the same amount of rabbit-derived IgG was to controls. 100 μL Protein G Sepharose (50%) was added to each sample and the agarose beads-antigen-antibody complex was collected. The complex was suspended in

100 μL 2×loading buffer and boiled for 5 min to release the antigen, antibody and Protein G beads. After centrifugation, the supernatant was subjected to electrophoresis and the whole protein lysate (Input), IgG eluted sample, and IP eluted sample were analyzed by western blotting. Finally, the protein signals were visualized using Pierce enhanced chemiluminescence reaction and exposed to medical X-ray film.

## Real-time reverse transcription-polymerase chain reaction (RT-PCR) analysis

Extraction of total RNA and reverse transcription was performed using extraction (TIANGEN, China) and reverse transcription (TaKaRa, Japan), as previously reported [13]. FS universal SYBR green master (Roche, Germany) was used for cDNA synthesis via a fluorescence-based quantitative PCR method and real-time PCR System (ABI, USA). RT was performed using 2.0 μl 5X gDNA Eraser Buffer, 1.0 μl gDNA Eraser, 1 μg Total RNA and RNase Free dH2O to a total volume of 10 μl. The reaction conditions were 42˚C for 2 min and rapidly cooled to 4˚C. Subsequently, 10.0 μl reaction solution, 1.0 μl PrimeScript RT Enzyme Mix 1, 1.0 μl RT Primer Mix, 4.0 μl 5X PrimeScript Buffer 2 and 4.0 μl RNase Free dH2O were used. The reaction conditions were 37˚C for 15 min, 85˚C for 5 sec, rapidly cooled to 4˚C and stored at -20˚C. A fluorescence-based qPCR method was performed using 2 μl cDNA, 10 μl SYBR Green (Takara Bio, Inc.), 0.6 μl PCR forward primer (10 μM), 0.6 μl PCR reverse primer (10 μM) and 6.8 μl dH2O in a 20 μl PCR reaction volume. The RT-qPCR reaction was run on a StepOne Real--Time PCR system (Applied Biosystems; Thermo Fisher Scientific, Inc.). The cycling parameters were as follows: Denaturing at 95˚C for 30 sec, 40 cycles at denaturing at 95˚C for 5 sec, primer annealing at 60˚C for 34 sec and extension temperature at 95˚C for 15 sec; final extension at 60˚C for 1 min and final denaturing at 95˚C for 15 sec. Gene expression levels were determined using the $2^{-\Delta\Delta Cq}$ method. Primers for GAPDH and FAK were from TaKaRa, Japan. GAPDH, Forward: 5′-GCA CCG TCA AGG CTG AGA AC-3′, Reverse: 5′-TGG TGA AGA CGC CAG TGGA-3′; FAK, Forward: 5′-CAA CCA CCT GGG CCA GTA TTA TC-3′, Reverse: 5′-CCA TAG CAG GCC ACA TGC TTTA-3′.

## Statistical analyses

SPSS 20.0 software was used for all statistical analyses and PCR results were analyzed by the Mann-Whitney U test. Differences between experimental groups and controls were assessed using the Student's t-test and analysis of variance (ANOVA), followed by LSD's test. A value of $p < 0.05$ was considered significant.

# Results

## Spheroid formation, cell migration and expression of FAK, p-FAK, AKT, p-AKT and stem cell marker proteins in HT29, HCT116, SW480, SW620 and RKO cells

Spheroid formation, cell migration and expression of FAK, p-FAK, AKT, p-AKT and stem cell markers in HT29, HCT116, SW480, SW620 and RKO cells were assessed to aid the selection of two suitable colorectal cancer cell-lines for subsequent experiments. The above five colorectal cancer cell lines had differing capacities for spheroid formation (Fig 1A). FAK, p-FAK, AKT, p-AKT, CD44, CD133, OCT4, Nanog and c-Myc proteins were expressed in all five cell-lines, although Sox2 was not. Expression of p-FAK and p-AKT was higher in HT29 and RKO than in HCT116, SW480 and SW620 cells (Fig 1B). In consideration of FAK or AKT being

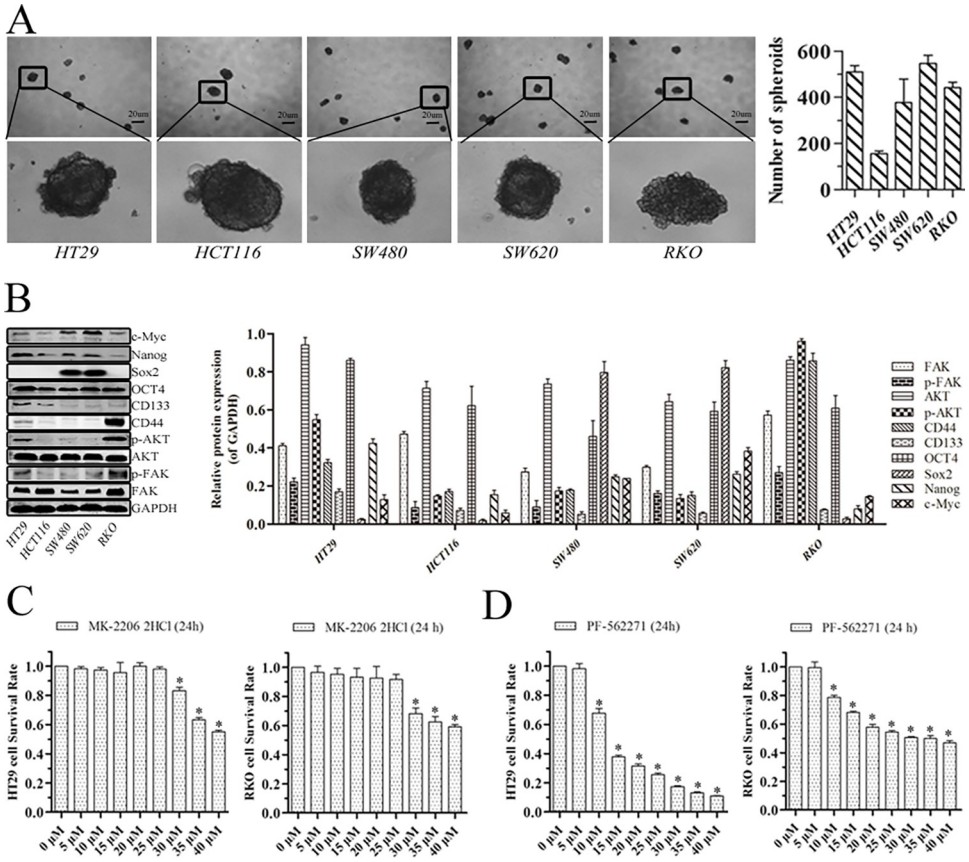

**Fig 1. Spheroid formation, cell migration, proteinexpression in colorectal cancer cells and the effect of MK-2206 2HCl and PF-562271 on survival of cells.** (A) Harvested cells were seeded under nonadherent culture conditions for spheroid formation (magnification, ×40; scale bar 20 μm). Numbers of spheroids were quantified after 7 days. (B) Protein levels assayed by western blotting. GAPDH was used as an internal control. (C) Viability of HT29 and RKO cells after treatment with MK-2206 2HCl. (D) Viability of HT29 and RKO cells after treatment with PF-562271. *p<0.05 compared with control group (0 μM).

manipulated by transfection experiments and specificinhibitors in subsequent experiments, HT29 and RKO cells were adopted.

## The effect of AKT inhibitor, MK-2206 2HCl, on survival of HT29 and RKO cells

On treatment with 25 uM MK-2206 2HCl for 24 h, viability of HT29 and RKO cells, measured by CCK8, was not significantly reduced (Fig 1C, p>0.05). At this time, MK-2206 2HCl could exert maximum efficacy in inhibiting target proteinexpression with minimal toxicity to thesurvival of HT29 or RKO cells.

## MK-2206 2HCl inhibited stem-like properties and migration of colorectal cancer cells

Cell immunocytochemistry analysis showed that expression of p-AKT, CD44 and c-Myc proteins was decreased by 24 h treatment with 25 uM MK-2206 2HCl for RKO cells (Fig 2A, p<0.05). In order to further clarify the impact of MK-2206 2HCl on the proteins expression of AKT, p-AKT and cancer stem cell markers, the proteins of HT29 and RKO cells were also

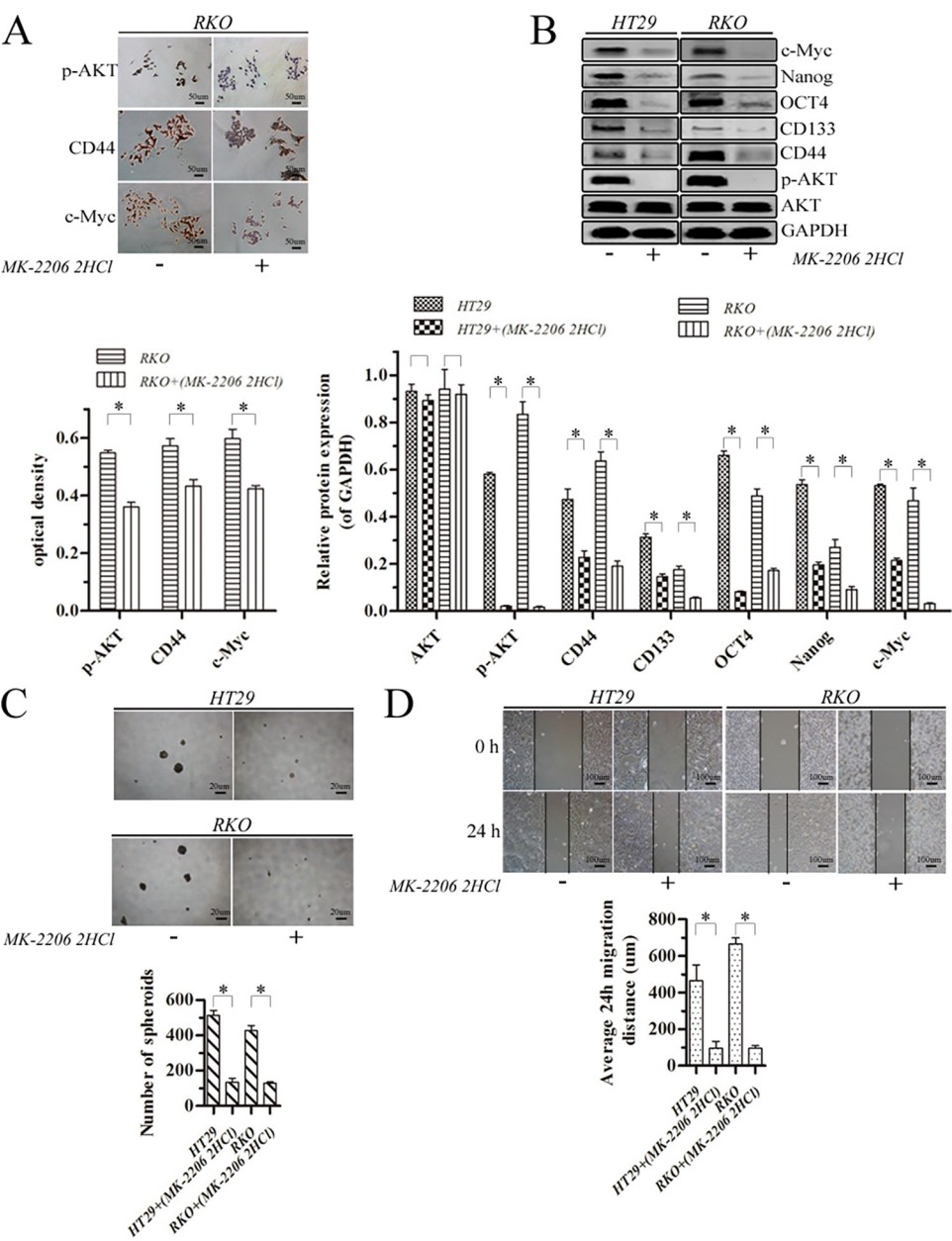

**Fig 2. MK-2206 2HCl inhibited stem-like properties and migration of HT29 and RKO cells.** (A) Protein levels were assayed by immunocytochemistry analysis (magnification, ×200; scale bar 50 μm). (B) Protein levels were assayed by western blotting. GAPDH was used as an internal control. (C) The harvested cells were seeded under nonadherent culture conditions for spheroid formation (magnification, ×40; scale bar 20 μm). Number of spheroids was quantified after 7 days. (D) Migratory potency of cells demonstrated by wound healing assays (magnification, ×100; scale bar 100 μm) and analysis. (*$p < 0.05$).

detected by western blotting. Treatment with 25 uM MK-2206 2HCl for 24 h reduced expression of p-AKT, CD44, CD133, OCT4, Nanog and c-Myc ($p < 0.05$) while that of AKT was unchanged ($p > 0.05$) in HT29 and RKO cells (Fig 2B). In the meantime, the spheroid forming ability was reduced (Fig 2C, $p < 0.05$), as was migration distance (Fig 2D, $p < 0.05$), indicating loss of migratory capacity. It has been shown that p-AKT regulated the expression of cancer stem cell markers previously [16]. The above findings show that MK-2206 2HCl reduced stem

cell-like properties and migration of colorectal cancer cells via suppression of p-AKT expression.

**The effect of FAK inhibitor, PF-562271, on survival of HT29 and RKO cells.** Treatment with 5 uM PF-562271 for 24 h did not reduce HT29 and RKO cell viability as measured by CCK8 (Fig 1D, p>0.05). At this time, PF-562271 could exert maximum efficacy in inhibiting target protein expression with minimal toxicity to the survival of HT29 or RKO cells.

## PF-562271 inhibited stem-like properties and migration of colorectal cancer cells though suppression of p-AKT expression

The RKO cells were treated with 5 uM PF-562271 for 24 h decreased p-FAK, p-AKT, CD44 and c-Myc expression by cell immunocytochemical analysis (Fig 3A, p<0.05) and expression of FAK, p-FAK, p-AKT, CD44, CD133, OCT4, Nanog and c-Myc by western blotting (p<0.05) in both HT29 and RKO cells while AKT was unchanged (p>0.05) (Fig 3B). In the meantime, HT29 and RKO spheroid formation was reduced (Fig 3C, p<0.05), as was migration distance (Fig 3D, p<0.05). The above findings supported that PF-562271 inhibited stem cell-like properties and migration of colorectal cancer cells perhaps via suppression of the FAK/p-FAK/p-AKT axis.

## Downregulation of FAK suppressed p-AKT expression and inhibited stem cell-like properties and migration of colorectal cancer cells

RKO cells were stably transfected with Lenti-FAK-EGFP-miR to reduce FAK expression and FAK mRNA was successfully suppressed (Fig 4A, p<0.05). Expression of FAK, p-FAK, p-AKT, CD44, CD133, OCT4, Nanog and c-Myc decreased (p<0.05) when FAK was downregulated and no change was seen in AKT expression (p>0.05) (Fig 4B, p<0.05). RKO cells after FAK knockdown [FAK(-)-RKO] showed reduced spheroid formation compared with transfected negative control (NC) and control (Con) RKO cells (Fig 4D, p<0.05). Moreover, migrational potency was significantly suppressed by FAK knockdown (Fig 4E, p<0.05).

## FAK regulated AKT phosphorylation in colorectal cancer cells

FAK and AKT proteins were shown to interact by co-immunoprecipation in RKO cells (Fig 4C). The RKO cells were treated with 5 uM PF-562271 for 24 h reduced p-AKT expression (p<0.05) but not that of AKT (p>0.05) (Fig 3B). Moreover, FAK knockdown in RKO cells reduced p-AKT expression (p<0.05) but not that of AKT (p>0.05) (Fig 4B). FAK appears to regulate AKT phosphorylation in colorectal cancer cells. In summary, the above findings show that the downregulation of FAK suppressed stem cell-like properties and migration of colorectal cancer cells partly by FAK modulation of AKT phosphorylation.

## Discussion

Cancer stem cells represent a population of cells within a tumor with highly tumorigenic properties and may be identified by the expression of cancer stem cell markers. Cancer stem-like cells make a major contribution to metastasis and are known to exist in a variety of solid tumors, including colorectal cancer [19,20]. Targeting cancer stem-like cells appears to be a promising approach for the elimination of cancer recurrence and improvement of clinical outcomes [21].

AKT is thought to be involved in stem cell survival. Activation of AKT, such as the action of PI3K leading to phosphorylation of AKT to p-AKT, increased the number of neural stem cells [22,23]. The AKT/Nanog pathway is critical for the maintenance of sarcoma cancer stem cells

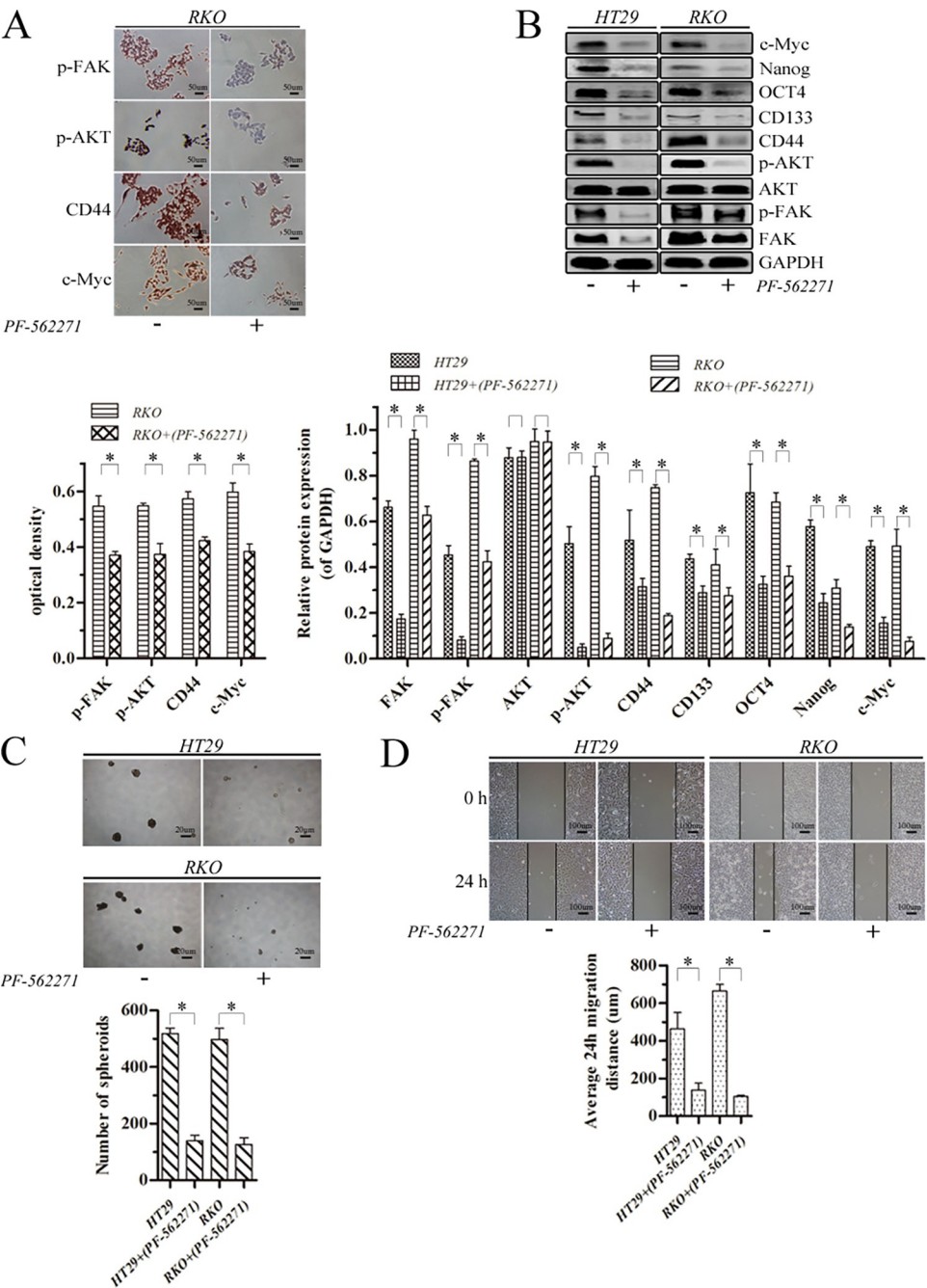

**Fig 3. PF-562271 inhibited stem-like properties and migration of HT29 and RKO cells.** (A) Protein levels were assayed by immunocytochemistry analysis (magnification, ×200; scale bar 50 μm). (B) Protein levels were assayed by western blotting. GAPDH was used as an internal control. (C) Harvested cells were seeded under nonadherent culture conditions for spheroid formation (magnification, ×40; scale bar 20 μm). Numbers of spheroids were quantified after 7 days. (D) Migratory potency of cells demonstrated by wound healing assays (magnification, ×100; scale bar 100 μm) and analysis. (*p<0.05).

and spheroid formation. Activation of the AKT/β-catenin pathway maintains the growth of cancer stem cells and directly modulates the expression of Nanog and OCT4 in colorectal cancer [24,25]. Inconsistently, some studies suggest that AKT is primarily induced in cancer stem

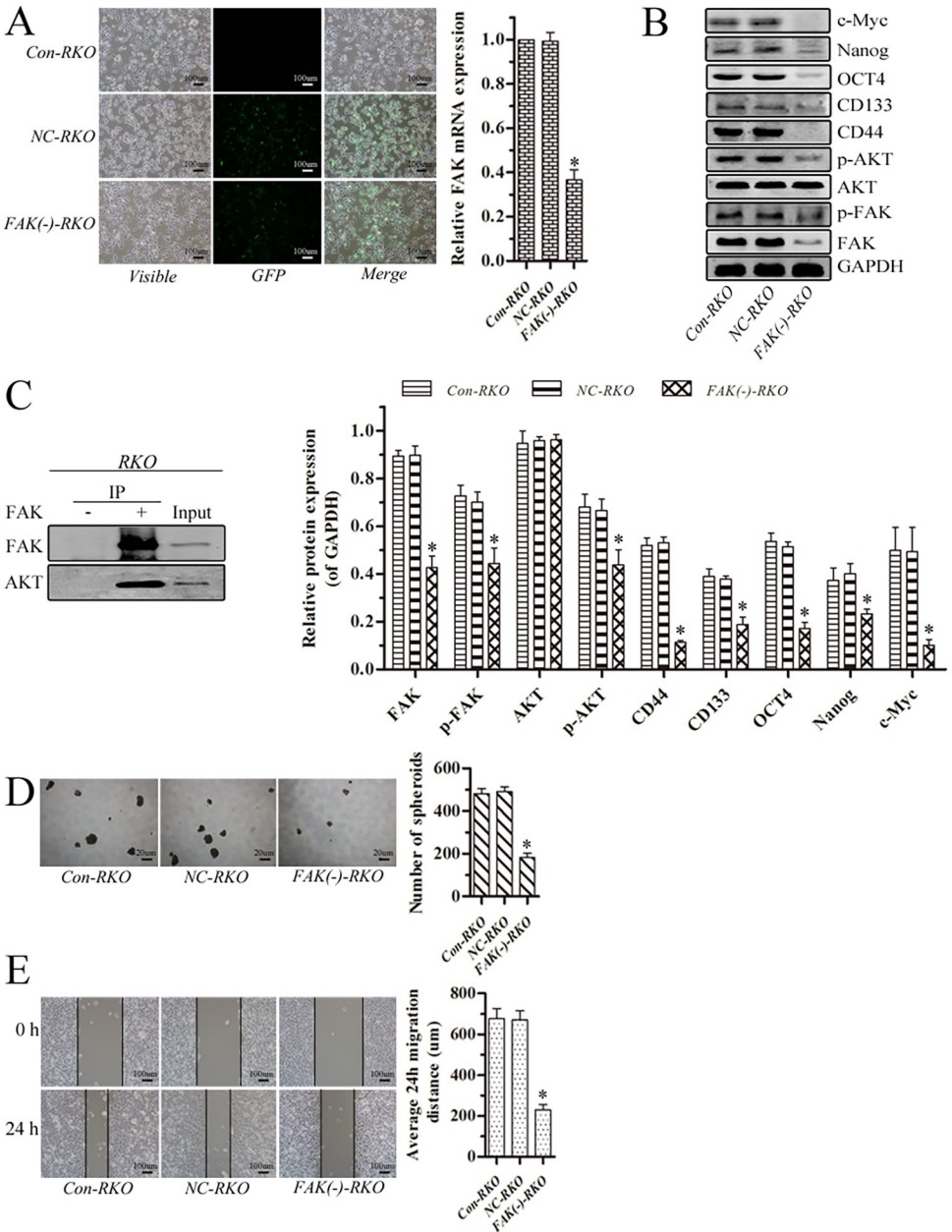

**Fig 4. Downregulation of FAK inhibited stem cell-like properties and migration of colorectal cancer cells through reducing AKT phosphorylation.** (A) Fluorescence microscopy of transfected and un-transfected RKO cells (magnification, ×100; scale bar 100 μm). Bar graphs illustrating RT-PCR analysis of FAK mRNA. GAPDH was used as a reference and the Con-RKO was set to 1. (B) Protein levels were assayed by western blotting. GAPDH was used as an internal control. (C) Interaction of FAK and AKT proteins in colorectal cancer cells shown by co-immunoprecipitation. (D) Harvested cells were seeded under nonadherent culture conditions for spheroid formation (magnification, ×40; scale bar 20 μm). Numbers of spheroids were quantified after 7 days. (E) Migratory potency of cells demonstrated by wound healing assays (magnification, ×100; scale bar 100 μm) and analysis. (*p<0.05).

cells through upregulation of the upstream Nanog [26]. A cross-regulation between AKT and Nanog may play a significant role in maintaining stem cell-like properties in tumors. In addition, AKT activation underpinned the stemness of colorectal cancer stem cells and CD133 expression changed when AKT was inhibited [23]. Increased CD44 expression in colorectal

cancer cells was shown to be linked to increased AKT expression by the superimposition of targeted proteomic analysis on transcriptomic analysis [27,28]. C-Myc also was a downstream molecule in the AKT pathway and siRNA knockdown of c-Myc effectively inhibited cancer stem cell-like features [29]. In the current study, low expression of p-AKT was associated with decreased expression of stem cell markers, Nanog, OCT4, CD133, CD44 and c-Myc, in RKO and HT29 cells. Spheroid formation was suppressed by AKT inactivation. These data indicated colorectal cancer cells losed stem cell-like properties partly by inactivation of AKT with reduced phosphorylation of AKT.

FAK, which is implicated in tumor growth and metastasis, traffic to and is present in the nucleus, affecting gene regulation in a kinase-independent manner. The anti-tumor activity of FAK inhibitors has been used to reinforce traditional cytotoxic chemotherapy and immuno-therapy in the form of combination treatments. PI3K activates AKT by phosphorylating serine 473 (Ser473) and the FAK/PI3K/AKT pathway is known to regulate cell motility. FAK was involved in the regulation of cell mobility via activation of the PI3K /AKT pathway, and associated phosphorylation of p85 subunits of tyrosine of PI3K in human cancer cells [28,30]. FAK and AKT proteins were shown to interact verified by co-immunoprecipitation and inhibition of FAK reduced p-AKT expression in this study. These data suggest a role for FAK in the regulation of AKT phosphorylation in colorectal cancer cells. FAK inhibition decreases cancer stem-like properties. FAK is implicated in modulating the resistant and aggressive phenotype of cancer stem cells [31]. The present study shows that FAK inhibitor decreased the expression of p-AKT and of stemness-related proteins, such as Nanog, OCT4, CD133, CD44 and c-Myc, in RKO and HT29 cells. When FAK expression of the transfected RKO cells was suppressed, levels of p-AKT and stemness markers were also decreased. FAK inhibition suppressed spheroid formation and migration of colorectal cancer cells.

In conclusion, the current study demonstrated that downregulation of FAK inhibited stem cell-like properties and migration of colorectal cancer cells partly due to altered modulation of AKT phosphorylation by FAK. Besides, pluripotency-associated factor Nanog expression was significantly attenuated by ERK knockdown [32]. FAK regulates cancer cell migration and stemness via ERK/Nanog pathway [33]. FAK is a potential tumor-suppressor target for colorectal cancer treatment, leading to reduced stem cells. Colorectal cancer stem-like cells are considered to be a primary cause of tumor recurrence. Elimination of cells with stem-like properties would make a significant contribution to improving clinical outcomes in patients with colorectal cancer.

## Supporting information

**S1 Raw images.**
(TIF)

## Acknowledgments

The authors would like to express their gratitude to EditSprings (https://www.editsprings.cn) for the expert linguistic services provided.

## Author Contributions

**Conceptualization:** Chunyan Xu.

**Data curation:** Chunyan Xu.

**Formal analysis:** Chunyan Xu.

**Funding acquisition:** Chunyan Xu.

**Methodology:** Chunyan Xu.

**Project administration:** Chunyan Xu, Wenlu Zhang, Chengxia Liu.

**Supervision:** Wenlu Zhang, Chengxia Liu.

**Validation:** Wenlu Zhang, Chengxia Liu.

**Writing – original draft:** Chunyan Xu.

**Writing – review & editing:** Chunyan Xu.

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
