## [Decision Letter · Decision Letter 0]

19 Jan 2023

PONE-D-22-25987FAK downregulation suppresses stem-like properties and migration of human colorectal cancer cellsPLOS ONE

Dear Dr. Xu,

Thank you for submitting your manuscript to PLOS ONE. After careful consideration, we feel that it has merit but does not fully meet PLOS ONE’s publication criteria as it currently stands. Therefore, we invite you to submit a revised version of the manuscript that addresses the points raised during the review process.

As you can see both reviewers have raised substantial issues which need to be addressed if the authors plan to resubmit this manuscript.

We look forward to receiving your revised manuscript.

Kind regards,

Salvatore V Pizzo

Academic Editor

PLOS ONE

Journal Requirements:

4. Thank you for stating the following in the Acknowledgments/Funding Section of your manuscript: 

This Project was supported by the Medical and Health Science and Technology Development Program of Shandong Province (No. 2019WS331).

Reviewers' comments:

Reviewer's Responses to Questions

**Comments to the Author**

1. Is the manuscript technically sound, and do the data support the conclusions?

Reviewer #1: Yes

Reviewer #2: Partly

2. Has the statistical analysis been performed appropriately and rigorously? 

Reviewer #1: Yes

Reviewer #2: Yes

3. Have the authors made all data underlying the findings in their manuscript fully available?

Reviewer #1: Yes

Reviewer #2: Yes

4. Is the manuscript presented in an intelligible fashion and written in standard English?

Reviewer #1: No

Reviewer #2: No

5. Review Comments to the Author

Reviewer #1: Dear editor,

The manuscript entitled ‘FAK downregulation suppresses stem-like properties and migration of human colorectal cancer cells’ is interesting and novel in the area of cancer. CSCs and their contribution to therapy resistance and tumor progression is without a doubt of value, and the area still needs investigations. The whole manuscript requires edition for grammar and writing errors in order to be a better draft for readers.

The sentence ‘cells protein was expression by western blotting analysis’ in the abstract section is wrong.

Please rewrite the Abstract in order to be more direct and persuasive. Sentences like ‘Cell viability was measured by cell counting kit-8 and cells protein was expression by western blotting analysis and immunocytochemistry’, ‘Co-expression of FAK and AKT in colorectal cancer cells was assessed by co-immunoprecipitation’, and ‘Spheroid forming ability was evaluated and cell migration measured by wound healing assays’ must not be included in the abstract section. Instead, please refer to the key findings of the study.

Introduction. The sentence ‘Both the EMT and the stem cell-like properties of tumor cells promote metastasis’ may pursue a wrong interpretation, as it has known EMT is a feature of cancer stemness not a separate event (10.1007/s12032-022-01801-0).

AKT is primarily induced in CSCs through upregulation of the upstream Nanog (J. Clin. Investig. 122 (11) (2012) 4077–4093) or PI3K (Proc. Natl. Acad. Sci. USA 110 (17) (2013) 6829–6834). Where is the place here for FAK?

Introduction. Please indicate here that CSCs are equipped with highly efficient DNA repair and effective redox tolerance systems in order to promote therapy resistance (10.1016/j.biopha.2022.113906), and their number is increased outstandingly in cancers with cold immunity (10.1002/jbt.22708).

Sincerely

Reviewer #2: The present study describes FAK regulation of stem cell-like properties of colon cancer cells.

FAK downregulation attenuates such stemness properties and cancer cell migration via AKT phosphorylation. FAK has been target of tumor-suppressor and downregulation of stem cells. The FAK science has well been documented to date. FAK in cytoplasmic region is present in cancer cells and p-AKT is upregulated in EMT, hepatoma and colon cancer metastasis. FAK/AKT/PI3K in EMT and stemness in cancer cells is also well known.

The present works are not justified for generalization, indicating a premature submission.

1. More mechanistic explanation in FAK interaction with its partner molecules should be gel-shifted using immuno-precipitation.

2. Several colorectal cancer cells have been analyzed in Fig.1 and etc. The genetic differences in p53 gene are characteristic and the FACK regulated is controlled by p53 and related genes. Therefore, the phenotype properties should be separately examined.

3. PI3K/PTEn coregulation is missed

The present study is not acceptable in its present form

6. PLOS authors have the option to publish the peer review history of their article (what does this mean?). If published, this will include your full peer review and any attached files.

Reviewer #1: No

Reviewer #2: **Yes: **Cheorl-Ho Kim

---

## [Author Response · Author response to Decision Letter 0]

30 Jan 2023

Response to Reviewer #1: Dear reviewer, thank you for your suggestion.

1.The grammar and writing errors of the manuscript have been revised and obtained a language certificate from EditSprings.

2.We have rewritten the Abstract. I hope it could meet your requirement.

3.We have rewritten the sentence "During the initiation of metastasis, epithelial cancer cells undergo an epithelial-mesenchymal transition (EMT), and as a result become migratory and invasive mesenchymal-like cells while acquiring stem cell-like properties and therapy resistance. It was known EMT is a feature of stem cell-like properties in tumor"in Introduction.

4.The AKT/Nanog pathway is critical for maintenance of sarcoma cancer stem cells and spheroid formation. Activation of AKT/β-catenin pathway maintains the growth of cancer stem cells and directly modulates the expression of Nanog and OCT4 in colorectal cancer (doi: 10.1038/s41389-020-00300-z.; doi:10.3892/ijmm.2021.4884). Inconsistently, some studies suggest that AKT is primarily induced in cancer stem cells through upregulation of the upstream Nanog (J. Clin. Investig. 122 (11) (2012) 4077–4093). A cross-regulation Maybe between AKT and Nanog, plays a significant role in maintains stem cell-like properties in tumor. FAK inhibition can suppress ovarian cancer cells migration and invasion through inhibiting downstream PI3K/AKT signaling. (doi: 10.1002/cbin.10184.)

5.We have added the sentence "Cancer stem cells are equipped with highly efficient DNA repair and effective redox tolerance systems in order to promote therapy resistance, and their number is increased outstandingly in cancers with cold immunity" in Introduction.

Response to Reviewer #2: Dear reviewer, thank you for your suggestion.

1.In our research, FAK and AKT protein was shown to interact indeed verified by co-immunoprecipitation.

2.In Fig 1, we show the capacity of spheroid formation, cell migration and expression of FAK, p-FAK, AKT, p-AKT, CD44, CD133, OCT4, Nanog and c-Myc in HT29, HCT116, SW480, SW620 and RKO colorectal cancer cells. Expression of p-FAK and p-AKT was higher in HT29 and RKO. In consideration of FAK or AKT was manipulated by transfection experiments and specific inhibitor in subsequent experiments, HT29 and RKO cells were adopted. The results shown in Fig.1 could explain why HT29 and RKO were selected for the subsequent experiments, so the phenotype properties may be dispensable for separately examination.

3.Some studies have reported that the FAK was involved in the regulation of cell mobility via activation of PI3K /AKT pathway, and associated phosphorylation of p85 subunits of tyrosine of PI3K in human cancer cells. (doi:10.1038/nrc3792) ( doi:10.18632/oncotarget.6399).

---

## [Decision Letter · Decision Letter 1]

21 Mar 2023

PONE-D-22-25987R1FAK downregulation suppresses stem-like properties and migration of human colorectal cancer cellsPLOS ONE

Dear Dr. Xu,

Thank you for submitting your manuscript to PLOS ONE. After careful consideration, we feel that it has merit but does not fully meet PLOS ONE’s publication criteria as it currently stands. Therefore, we invite you to submit a revised version of the manuscript that addresses the points raised during the review process.

A number of issues were raised by one of the reviewers which should be considered if the authors plan to submit a revised manuscript.

We look forward to receiving your revised manuscript.

Kind regards,

Salvatore V Pizzo

Academic Editor

PLOS ONE

Reviewers' comments:

Reviewer's Responses to Questions

**Comments to the Author**

1. If the authors have adequately addressed your comments raised in a previous round of review and you feel that this manuscript is now acceptable for publication, you may indicate that here to bypass the “Comments to the Author” section, enter your conflict of interest statement in the “Confidential to Editor” section, and submit your "Accept" recommendation.

Reviewer #1: All comments have been addressed

Reviewer #3: All comments have been addressed

2. Is the manuscript technically sound, and do the data support the conclusions?

Reviewer #1: Yes

Reviewer #3: Yes

3. Has the statistical analysis been performed appropriately and rigorously? 

Reviewer #1: Yes

Reviewer #3: Yes

4. Have the authors made all data underlying the findings in their manuscript fully available?

Reviewer #1: Yes

Reviewer #3: Yes

5. Is the manuscript presented in an intelligible fashion and written in standard English?

Reviewer #1: Yes

Reviewer #3: Yes

6. Review Comments to the Author

Reviewer #1: The manuscript is acceptable in the current form. All the required amendments are addressed precisely

Reviewer #3: - Please carefully check the English grammar, which has many errors and typos.

- In Real time-PCR method, what is the thermal cycling conditions? How to calculate and analyze the level of gene expression? These should be specified in the method.

- In the results section, the authors have to explain the detail of each result more.

- All images of all figures have not the scale bar, it needs to show in the image and also specify in the figure legends.

- The authors need to link and interpret the data in the results and discussion about the two different cell lines, HT29 and RKO.

7. PLOS authors have the option to publish the peer review history of their article (what does this mean?). If published, this will include your full peer review and any attached files.

Reviewer #1: No

Reviewer #3: No

---

## [Author Response · Author response to Decision Letter 1]

28 Mar 2023

Dear reviewers,

 Thank you for your letter and for the reviewer’ comments concerning our manuscript. Those comments are all valuable and very helpful for revising and improving our paper. We have studied comments carefully and have made correction which we hope meet with approval.

Reviewer #3:

1. Please carefully check the English grammar, which has many errors and typos.

 We invited the high qualified and native English speakers at EditSprings again to modify the English grammar in our manuscript.

2. In Real time-PCR method, what is the thermal cycling conditions? How to calculate and analyze the level of gene expression? These should be specified in the method.

 We added the thermal cycling conditions and the formula to calculate and analyze the level of gene expression in the method section.

3. In the results section, the authors have to explain the detail of each result more.

 We made some changes in the results section. We hope it could meet your requirement.

4. All images of all figures have not the scale bar, it needs to show in the image and also specify in the figure legends.

 We added the scale bar in all images of all figures. I hope it could meet your requirement.

5. The authors need to link and interpret the data in the results and discussion about the two different cell lines, HT29 and RKO.

 In the discussion section, we have linked and interpreted the data of the results. HT29 and RKO are all colorectal cancer cell lines, so we did not discuss them separately. 

 We sincerely hope that this revised manuscript has addressed all your comments and suggestions. We appreciated for reviewer’ warm work earnestly, and hope that the correction will meet with approval. Once again, thank you very much for your comments and suggestions.

---

## [Decision Letter · Decision Letter 2]

5 Apr 2023

PONE-D-22-25987R2FAK downregulation suppresses stem-like properties and migration of human colorectal cancer cellsPLOS ONE

Dear Dr. Xu,

Thank you for submitting your manuscript to PLOS ONE. After careful consideration, we feel that it has merit but does not fully meet PLOS ONE’s publication criteria as it currently stands. Therefore, we invite you to submit a revised version of the manuscript that addresses the points raised during the review process.

 Your manuscript is basically acceptable for publication but one reviewer has indicated a labelling error. It is preferable for the authors to correct this if the reviewer is correct rather than editorial change of the legend.

We look forward to receiving your revised manuscript.

Kind regards,

Salvatore V Pizzo

Academic Editor

PLOS ONE

Journal Requirements:

Reviewers' comments:

Reviewer's Responses to Questions

**Comments to the Author**

1. If the authors have adequately addressed your comments raised in a previous round of review and you feel that this manuscript is now acceptable for publication, you may indicate that here to bypass the “Comments to the Author” section, enter your conflict of interest statement in the “Confidential to Editor” section, and submit your "Accept" recommendation.

Reviewer #3: All comments have been addressed

2. Is the manuscript technically sound, and do the data support the conclusions?

Reviewer #3: Yes

3. Has the statistical analysis been performed appropriately and rigorously? 

Reviewer #3: Yes

4. Have the authors made all data underlying the findings in their manuscript fully available?

Reviewer #3: Yes

5. Is the manuscript presented in an intelligible fashion and written in standard English?

Reviewer #3: Yes

6. Review Comments to the Author

Reviewer #3: All images of all figures have already added the scale bar, but in the figure legends are not correct. Authors only wrote the word "(magnification, ×40)", so it should be written "(magnification, ×40; scale bar 20 μm)". Please edit and rewrite them all of the figure legends.

7. PLOS authors have the option to publish the peer review history of their article (what does this mean?). If published, this will include your full peer review and any attached files.

Reviewer #3: No

---

## [Author Response · Author response to Decision Letter 2]

5 Apr 2023

Dear reviewer,

 Thank you for your letter and for the reviewer’ comments concerning our manuscript. This comment is valuable and very helpful for revising and improving our paper. We have studied the comment carefully and have made correction which we hope meet with approval.

Reviewer #3:

All images of all figures have already added the scale bar, but in the figure legends are not correct. Authors only wrote the word "(magnification, ×40)", so it should be written "(magnification, ×40; scale bar 20 μm)". Please edit and rewrite them all of the figure legends.

 We edit and rewrite the figure legends as your suggestion. I hope it could meet your requirement.

 We sincerely hope that this revised manuscript has addressed all your comments and suggestions. We appreciated for reviewer’ warm work earnestly, and hope that the correction will meet with approval. Once again, thank you very much for your comments and suggestions.

---

## [Decision Letter · Decision Letter 3]

11 Apr 2023

FAK downregulation suppresses stem-like properties and migration of human colorectal cancer cells

PONE-D-22-25987R3

Dear Dr. Xu,

We’re pleased to inform you that your manuscript has been judged scientifically suitable for publication and will be formally accepted for publication once it meets all outstanding technical requirements.

Kind regards,

Salvatore V Pizzo

Academic Editor

PLOS ONE

Additional Editor Comments (optional):

Reviewers' comments:

Reviewer's Responses to Questions

**Comments to the Author**

1. If the authors have adequately addressed your comments raised in a previous round of review and you feel that this manuscript is now acceptable for publication, you may indicate that here to bypass the “Comments to the Author” section, enter your conflict of interest statement in the “Confidential to Editor” section, and submit your "Accept" recommendation.

Reviewer #3: All comments have been addressed

2. Is the manuscript technically sound, and do the data support the conclusions?

Reviewer #3: Yes

3. Has the statistical analysis been performed appropriately and rigorously? 

Reviewer #3: Yes

4. Have the authors made all data underlying the findings in their manuscript fully available?

Reviewer #3: Yes

5. Is the manuscript presented in an intelligible fashion and written in standard English?

Reviewer #3: Yes

6. Review Comments to the Author

Reviewer #3: (No Response)

7. PLOS authors have the option to publish the peer review history of their article (what does this mean?). If published, this will include your full peer review and any attached files.

Reviewer #3: No

---

## [Editor Report · Acceptance letter]

13 Apr 2023

PONE-D-22-25987R3 

FAK downregulation suppresses stem-like properties and migration of human colorectal cancer cells 

Dear Dr. Xu:

I'm pleased to inform you that your manuscript has been deemed suitable for publication in PLOS ONE. Congratulations! Your manuscript is now with our production department. 

Kind regards, 

on behalf of

Dr. Salvatore V Pizzo 

Academic Editor

PLOS ONE